



# Refactoring the EVP solver for improved performance - a case study based on CICE v6.5

Till Andreas Soya Rasmussen[1], Jacob Poulsen[2], Mads Hvid Ribergaard[1], Ruchira Sasanka[2], Anthony P. Craig[4], Elizabeth C. Hunke[3], and Stefan Rethmeier[1]

[1]Danish Meteorological Institute, Sankt Kjelds Plads 11, 2100 Copenhagen, Denmark
[2]Intel Corporation
[3]MS-B216, Los Alamos National Laboratory, Los Alamos, NM, 87545, USA
[4]Contractor to Science and Technology Corporation, Seattle, WA

**Correspondence:** Till Andreas Soya Rasmussen (tar@dmi.dk)

**Abstract.** This study focuses on the performance of CICE and its Elastic-Viscous-Plastic (EVP) dynamical solver. The study has been conducted in two steps. First, the standard EVP solver has been extracted from CICE for experiments with refactored versions of it. Secondly, one refactored version was integrated and tested as part of the full model. Two dominant bottlenecks were revealed. The first is the number of MPI and OpenMP synchronization points required for halo exchanges during each

time-step combined with the irregular domain of active sea ice points. The second is the lack of Single Instruction Multiple Data (SIMD) code generation.

The study refactors the standard EVP solver based on two generic patterns. The first pattern exposes how general finite-differences on masked multi-dimensional arrays can be expressed in order to produce significantly better code generation. The primary change is that the memory access pattern is changed from random access to direct access. The second pattern exposes

an alternative approach to handle static grid properties.

The measured single core improvement is increased by more than a factor of five compared to the standard implementation. The refactored implementation strong scales on the Intel® Xeon® Scalable Processor Series node until the available bandwidth of the node is used. For the Intel® Xeon® CPU Max Series Series there is sufficient bandwidth to allow the strong scaling to continue for all the cores on the node resulting in a single node improvement factor of 35 over the standard implementation.

This study also show improved performance on GPU processors.

## 1 Introduction

Numerical models of the Earth system and its components (e.g. ocean, atmosphere and sea ice) rely heavily on high performance computers (HPC) (Lynch, 2006). When the first massively parallel computers emerged, steadily increasing CPU speeds improved performance sufficiently to support faster time-to-solution, higher resolution and improved physics. However, over

the last decade improved performance originates from an ever increasing number of cores, each supporting an increasing number of SIMD lanes. Thus, for codes to run efficiently on today's hardware, they need to have excellent support of threads and efficient SIMD code generation.





Earth System Model (ESM) implementations often use static grid properties that are computed once and either carried from one subroutine to the next in a data structures or accessed as global data structures. From a logical perspective it makes sense to not recompute the same thing over and over. Historically, floating point operations have been expensive and therefore this also made sense from a compute performance perspective. Modern hardware has changed and memory storage, particularly the bandwidth to memory, is the scarce resource. Compared with floating point operations, it is not only scarce but also far more energy demanding. ESM components must be refactored to adapt to modern hardware features and limitations.

This study focuses on the dynamical solver of CICE (Hunke et al., 2023a) a sea ice component in ESMs. In general, the same sea ice models are used both in climate models and in operational systems with different settings (Hunke et al., 2020). The dynamic solver of sea ice models is physically important as it calculates the momentum equation including internal sea ice stresses. The dynamics are usually based on the viscous-plastic (VP) model developed by Hibler (1979), which has a singularity that is difficult for numerical solvers to handle. Consequently, the Elastic-Viscous-Plastic approach was developed by Hunke and Dukowicz (1997). This is designed to solve the nonlinear VP equations using parallel computing architectures. In order to achieve a fast and non-singular solution, elastic waves were added to the VP solution. The EVP solution ideally converges to the VP solution via hundreds of EVP iterations, which dampen the elastic waves. Bouillon et al. (2013) and Kimmritz et al. (2016) showed that the number of iterations to EVP convergence could be controlled and reduced, but this remains computationally expensive. Koldunov et al. (2019) reports that 550 iterations are needed in order to reach convergence with the traditional EVP solver in the finite element model FESOM2 at a resolution of 4.5km. According to Bouchat et al. (2022), the models of their inter-comparison used a wide range of number of EVP subcycles from 120 to 900. Thus while several approaches have been proposed to reduce the number of iterations, it is likely that some of these model systems do not iterate to convergence. The motivation for reducing the number of subcycles may be the performance in terms of time-to-solution.

The number of subcycles needed to converge depends on the application, the configuration and the resolution. Unfortunately, the dynamics is one of the most computationally burdensome parts of the sea ice model. As an example, timings have been measured for standalone sea ice simulations using the operational model at the Danish Meteorological Institute (DMI) for the 1st of March 2020. The fraction of the dynamical core of the total runtime increases from approximately 15% to approximately 75% as the number of subcycles increases from 100 to 1000. Similar timings with other seasons, domains and/or a different strategy for allocation of memory will change the fractions, but there will still be a significant increase when the number of subcycles are increased. This motivates a refactorization of this part of the model system.

Another challenge for sea ice components in global Earth system models is the load balancing across the domain, since sea ice covers only 12% and 7% of the ocean and Earth's surface, respectively (Weeks, 2010). In addition, the sea ice cover varies significantly in time and space, particularly with the season. This adds additional complexity, especially for regional setups as the number of active sea ice points varies. Craig et al. (2015) also discusses the inherent load imbalance issues and implements some advanced domain decompositions to improve the load balance in CICE.

This study demonstrates that the EVP model can be refactored to obtain a significant speedup and that the method is useful for the rest of the sea ice model as well as for other ESM components. Section 2 presents the standard EVP solver, the refactorization, the standalone test and the setup of the experiments. Section 3 analyzes the results, and section 4 provides a





discussion of the developments and next steps, including discussion of an improved integration of the refactored EVP solver into CICE. The integration demonstrated in this study focus on correctness alone. Section 5 summarizes the conclusions.

## 2 The EVP solver

The aim of this study is to optimize the EVP solver of CICE by refactoring the code. This section describes the analyses of the existing solver and improvements made in this study.

### 2.1 Standard implementation

The CICE grid is parallelized based on 2D blocks including halos that are required for the finite difference calculation within the EVP solver. Communication between the blocks is based on MPI and/or OpenMP. The EVP dynamics is calculated at every time step, but due to the nature of the finite differences it is necessary to update the velocities on the halo at every subcycling step. Input to the EVP dynamical core is external forcing from the ocean and atmosphere components and internally from sea ice conditions. Stresses and velocities are output for use elsewhere in the code, e.g. to calculate advection.

Listing 1 provides a schematic overview of the standard EVP algorithm, which consists of an outer convergence loop, two inner stages (`stress` and `stepu`), and a halo swap of the velocities. Each of the two inner stages carry an inner loop with its own trip-count based on subsets of the active points. The two inner stages within the subcycling exchange arrays used in the other stage. Therefore it is necessary to synchronize after each of the stages.

The inner stages operate on a subset of the grid points in 2D space. The grid points are classified into land points, water points and two types of active ice points, namely $U$ cell points and $T$ cell points. The active points are defined by a threshold of the sea ice concentration and the mass of sea ice and snow, and thus the location and the number of active points may change at every time step, although they remain constant during the subcycling. The sets of ice points are labeled $T$ cells and $U$ cells according to the Arakawa definition of the B-grid Arakawa and Lamb (1977). $T$ refers to the center cell and $U$ refers to the velocity points at corners. Most grid cells have both active $T$ and $U$ points, but there are points that only belong to one of these subgroups. For instance, there may be ice in the cell center ($T$) but the $U$ point lies along a coastline and is therefore inactive. The land and water points are static for a given configuration, and the number of active sea ice points are always a subset of the water points, changing in time during the simulation due to the change of season and external forcing.

**Listing 1.** A schematic view of the subcycling of the EVP algorithm

```
 1: do k = 1, ksub       ! ksub sub-cycles per model timestep
 2:    ! stage1 aka stress: use variables on T cells and velocities on U cells to
 3:    ! define stress* on T cells and stage-interface vectors
 4:    do i=1,nt          ! nt is number of active T cells at given timestep
 5:       ! Finite-Difference computations here
 6:       ...
 7:    enddo
 8:    ! stage2 aka stepu, use variables on U cells and stage-interface
```





```
 9:    ! to define new velocities* and new vars on U cells
10:    do j=1,nu        ! nu is number of active U cells at given timestep
11:       ! Finite-Difference computations here
12:       ...
13:    enddo
14:    ! data dependencies: references in stage1 are set in stage2
15:    ! halo_swap with OpenMP and MPI neighbors
16: enddo
```

From a workload perspective, the arithmetic in the EVP algorithm confines itself to short-latency operations: `add`, `mult`, `div` and `sqrt`. Despite the computations involved in EVP the computation intensity is 0.3 FLOP/Byte, which makes the workload highly bandwidth bound. Achieving a well-balanced representation of the workload is a huge challenge for any parallel algorithm working on these irregular sets, which has been recognized in earlier papers on CICE performance (Craig et al., 2015).

## 2.2 The refactored implementation

This section describes the refactored EVP solver. The refactored implementation was done in two steps. First we focused on improving the *core-level* parallelism and second on improving the *node-level* parallelism. The intention in the first step is to establish a solid single-core baseline before diving into the thread parallelism of the solver.

### 2.2.1 Single core refactorization

An important part of the refactorization is memory access patterns and how to reduce the memory-bandwidth pressure at the cost of additional floating point operations. Listing 2 shows a snippet of the standard code (`v0-base`) before the first refactoring. The challenge here is that any compiler will see the memory access pattern caused by the indirect addressing as random memory access and will consequently refrain from using modern vector (SIMD) instructions in its code generation.

Moreover, the code fragment reveals a classical finite-difference pattern that is similar to the refactorization pattern shown in chapter 3 of (Reinders and , Eds).

The new EVP solver introduces 1D structures instead of the original 2D structures and adds additional neighbor indexing overhead, thereby still tracking the neighboring cells required by the finite-difference scheme. This change allows the compiler to see the memory access pattern as mostly direct addressing with some indirect addressing required for accessing neighboring states in neighboring cells. The compiler will consequently be able to generate SIMD instructions for the loop and will handle the remaining indirect addressing with SIMD gather instructions. The ratio of indirect to direct addressing is 10%-20%, depending on which of the two EVP loops we refer to. The refactored code is shown in listing 3. For the Fortran programmer, the two fragments look almost identical but for the Fortran compiler the two fragments look very different, and the compiler will be able to convert the latter fragment straight into an efficient ISA representation. The change of data-structures from 2D





to 1D is the only difference between `v0-base` and `v1-simd`. The computational intensity of a loop iteration of `v0-base`
and `v1-simd` is identical.

**Listing 2.** Fragment showing Finite-Difference dependencies in the standard version of EVP (`v0-base`).

```
1: ! VERSION: v0-base
2: subroutine stress(...,nx,ny,icellt,indxti,...)
3:    real (kind=dbl_kind), intent(in), dimension(nx,ny) :: cyp, ...
4:    ...
5:    do ij = 1, icellt
6:       i = indxti(ij)
7:       j = indxtj(ij)
8:       ! smaller FD-block with column dependencies (i+-1,j+-1)
9:       divune    = cyp(i,j)*uvel(i   ,j  ) - dyt(i,j)*uvel(i-1,j  ) &
10:                  + cxp(i,j)*vvel(i   ,j  ) - dxt(i,j)*vvel(i   ,j-1)
11:      ...
12:      ! larger block with no column dependencies
13:      stressp_1(i,j) = (stressp_1(i,j) + c1ne*(divune - Deltane)) &
14:                       * denom1
15:      ...
16:   enddo
17: end subroutine stress
```

**Listing 3.** Fragment showing Finite-Difference dependencies in the refactored version of EVP (`v1-simd`).

```
1: ! VERSION: v1-simd
2: subroutine stress(...)
3:    real (kind=dbl_kind), dimension(:), intent(in), contiguous ::  uvel, ...
4:    ...
5:    do iw = il, iu
6:       tmp_uvel_ee = uvel(ee(iw))
7:       ...
8:       divune    = cyp(iw)*uvel(iw) - dyt(iw)*tmp_uvel_ee                    &
9:                  + cxp(iw)*vvel(iw) - dxt(iw)*tmp_vvel_se
10:      ...
11:      stressp_1(iw) = (stressp_1(iw) + c1ne*(divune - Deltane)) * denom1
12:      ...
13:   enddo
14: end subroutine stress
```





CICE utilizes static grid properties which are computed once at the initialization step and then re-used in the rest of the simulation. As discussed in the introduction, it makes sense to reduce the bandwidth pressure by adding more pressure on the floating point engines. This strategy adds floating point operation overhead but reduces memory storage, and more importantly, it reduces the memory bandwidth pressure. Our final refactored versions (v2*) have substituted 7 static grid arrays with 4

static base arrays plus some run time computations of local scalars, deriving all 7 original arrays as local scalars, cf. listing 4 that shows how the arrays cxp and cyp are derived. The v2* versions are further discussed in section 2.2.2.

**Listing 4.** Fragment showing Finite-Difference dependencies in the refactored version of EVP (v2).

```
1: ! VERSION: v2-simd
2: subroutine stress(...)
3:    real(kind=dbl_kind), dimension(:), intent(in), contiguous :: uvel, ...
4:    ...
5:    do iw = il, iu
6:       tmp_uvel_ee = uvel(ee(iw))
7:       tmp_cxp    = c1p5 * htn(iw) - p5 * htnm1(iw) ! derive cxp from htn, htnm1
8:       tmp_cyp    = c1p5 * hte(iw) - p5 * htem1(iw) ! derive cyp from htn, htnm1
9:       ...
10:      divune     = tmp_cyp*uvel(iw) - dyt(iw)*tmp_uvel_ee                    &
11:                 + tmp_cxp*vvel(iw) - dxt(iw)*tmp_vvel_se
12:      ...
13:   enddo
14: end subroutine stress
```

### 2.2.2 Single node refactoring - OpenMP and OpenMP target

The existing standard parallelization operates on blocks of active points and elegantly uses the same parallelism for OpenMP

and MPI giving users flexibility to run hybrid. It allows for a number of different methods to distribute the blocks onto the compute units to meet the complexity of the varying sea ice cover. There is no support for GPU offloading in the standard parallelization. The refactored EVP kernel demonstrates support for GPU offloading and showcases how this can be done in a portable fashion, confining ourselves to open standards. The underlying idea in the OpenMP parallelization is two-fold. First, we want to make the OpenMP synchronization points significantly more light-weight than what is found in the current standard

implementation. The OpenMP synchronization points in the existing implementation involves explicit memory communication of data in the blocks used in the parallelization. The proposed OpenMP synchronization only requires an OpenMP barrier to ensure cache coherency and no explicit data movement. Second, we want to improve the granularity of the parallelization unit which will allow us to balance the workload better when running with higher number of cores and hence allow for better scaling. The granularity will change from blocks of active points into single active points.





Keeping the dependencies between the two stages in mind, we have two inner loops that can be parallelized. We must take into account that the $T$-cells and $U$-cells may not be identical sets, and we cannot even assume that one is included in the other. However, it is fair to assume that the difference between the $T$ and $U$ sets of active points is negligible and instead of treating the two sets as totally independent, we take advantage of their large overlap. It will help the performance tremendously both in terms of cache and in terms of None-Uniform Memory access (NUMA) placement to treat the two loops with the same trip

count. This requires some additional overhead in the code in order to skip the inactive points in the $T$ loop and in the $U$ loop.

    The OpenMP standard published by OpenMP Architecture Review Board (2021) provides several options, and later Fortran standards (Fortran-2008 and Fortran-2018) also provide an opportunity to express this parallelism purely within Fortran. For OpenMP, we can do this either in an *outlined* fashion (see listing 5) or in an *inlined* fashion (see listing 6).

**Listing 5.** Fragment showing outlined OpenMP parallelization of EVP (`omp-outline`).

```
1:   ! VERSION: v2-omp-outline
2:   do i = 1, ndte
3:     !$omp parallel do schedule(runtime) private(iw)
4:     do iw = 1, union_tripcount
5:       call stress(iw,...)
6:     enddo
7:     !$omp end parallel do
8:     !$omp parallel do schedule(runtime) private(iw)
9:     do iw = 1, union_tripcount
10:      call stepu(iw,...)
11:    enddo
12:    !$omp end parallel do
13:  enddo
```

**Listing 6.** Fragment showing traditional inlined (and OpenMP offloadling) OpenMP parallelization of EVP (`omp-inline`).

```
1:   ! VERSION: v2-omp-inline
2:   subroutine stress(...)
3:     ...
4: #ifdef _OPENMP_TARGET
5:     !$omp target teams distribute parallel do
6: #else
7:     !$omp parallel do schedule(runtime) default(none) shared(...) private(...)
8: #endif
9:     do iw = lb, ub
10:      if (skipt(iw)) cycle
11:      ...
12:    enddo
```




```
13:   end subroutine stress
```

Alternatively, it can be done without dragging all the OpenMP runtime scheduling into the picture. OpenMP can be used as a
short-hand notation for handling the spawning of the threads and then manually do the loop-splitting, as illustrated in listing 7.
We refer to this approach as the Single Program Multiple Data (SPMD) approach (Reinders and , Eds; Levesque and Vose,
2017). In addition to its simplicity, it has the advantage that we can add balancing logic to the loop decomposition, accounting
for the application itself and its data. Finally, for the sake of completeness, we also parallelize the EVP solver using the newer
taskloop constructs available in OpenMP (see listing 8).

**Listing 7.** Fragment showing SPMD OpenMP parallelization of EVP (`omp-SPMD`).

```
1:    ! VERSION: v2-omp-SPMD
2:    !$omp parallel private(i)
3:    do i = 1, ndte
4:       call stress(union_tripcount, ...)
5:       !$omp barrier
6:       call stepu(union_tripcount, ...)
7:       !$omp barrier
8:    enddo
9:    !$omp end parallel
10:   ....
11:   subroutine stress(....)
12:      ...
13:      call domp_get_domain(lb,ub,il,iu) ! get local thread bounds il, iu
14:      do iw = il, iu
15:         if (skipme(iw)) cycle
16:         ....
17:      enddo
18:   end subroutine stress
```

**Listing 8.** Fragment showing the newer OpenMP taskloop parallelization of EVP (`omp-taskloop`).

```
1:    ! VERSION: v2-omp-taskloop
2:    subroutine stress(...)
3:       ...
4:       !$omp parallel                              &
5:       !$omp single                                &
6:       !$omp taskloop simd                         &
7:       !$omp default(none) private(...) shared(...)  &
8:       do iw = lb, ub
9:          if (skipt(iw)) cycle
```





```
270   10:         ...
      11:       enddo
      12:       !$omp end taskloop simd
      13:       !$omp end single
      14:       !$omp end parallel
275   15:    end subroutine stress
```

A pure Fortran approach is available with the newer `do concurrent` (see listing 9) and here we may use compiler options to target either the CPU or GPU offloading.

**Listing 9.** Fragment showing pure Fortran-2018 approach to parallelism (`fortran-2018`).

```
280   1:    ! VERSION: v2-fortran-2018
      2:    subroutine stress (...)
      3:       ...
      4:       do concurrent (iw=lb:ub) DEFAULT(NONE)                        &
      5:                              SHARED(ee, ne, se, skipme, strength, uvel, vvel,   &
285   6:                                     ...                            &
      7:                              LOCAL(divune, divunw, divuse, divusw,   &
      8:                                     ...                            &
      9:       if (skipme(iw)) cycle
      10:       ...
290   11:       enddo
      12:    end subroutine stress
```

### 2.3 Test cases

Three domains have been used to test the refactorization of the EVP solver, shown in figure 1; the number of grid points and
295  their classification are listed in table 1.

Figure 1 (a) is the operational sea ice model domain at DMI (Ponsoni et al., 2023), covering the pan-Arctic area. The Regional Arctic System Model (RASM) domain in Figure 1 (b) (Brunke et al., 2018) also covers the pan-Arctic area. These two systems are computationally expensive and are therefore used to analyze, demonstrate and test the performance of the new EVP solver. Tests are based on restart files from both winter (March 1) and summer (September 1), where the ice reaches close
300  to its largest and the smallest extents. All performance results shown in this manuscript originate from the RASM domain, which has the most grid points. Neither of these model systems include boundary conditions, which simplifies the problem. The gx1 (1°) domain shown in figure 1(c and d) is included to test correctness and updates of the cyclic boundary conditions. In addition, gx1 is used for testing the numerical noise level for long runs using optimized flags within the CICE test suite.

Figure 1 indicates and table 1 highlights that most active points are both $T$ and $U$ points. This confirms the assumption in
305  the design of the refactored EVP solver. However there are cases where a grid point can be either a $T$ or a $U$ grid point. It is



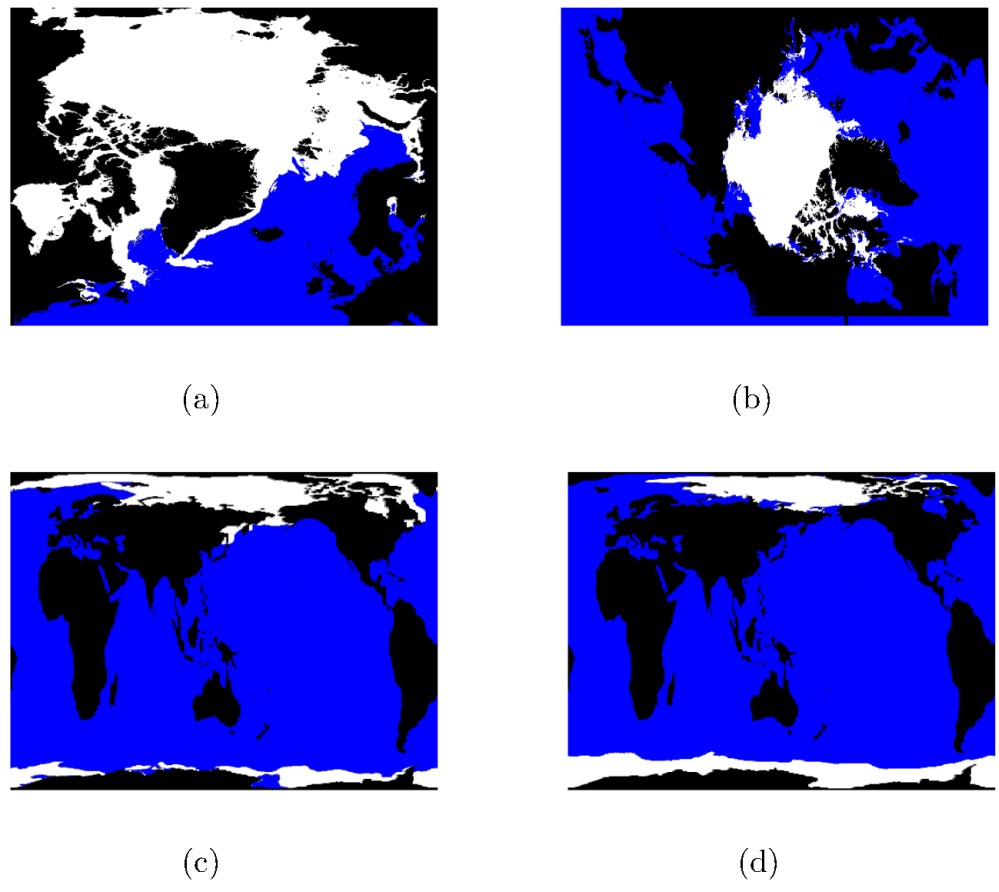

**Figure 1.** Three domains are used to test the new EVP solver. One restart from each of the regional domains are shown and one winter and one summer restart are shown for the global domain. a) DMI domain on the first of March 2020; b) RASM domain on the first of September 2018; c) gx1 domain on the first of March 2005; d) gx1 domain on the first of September 2005. Black is land, blue is ocean, gray points are either active $T$ or $U$ points and white points are active in both the $T$ and the $U$ points.

clear from figure 1 that represents the ice cover in a winter and a summer situation that the variation of active sea ice points is significant. Table 1 quantifies the variation of the different categories of grid cells.

The number of active grid cells varies for the two pan-Arctic domains and are approximately half in summer when compared to winter. The minimum and maximum sea ice extent is expected mid-September and mid-March respectively. A sinusoidal variation is expected in the period between the minimum and the maximum for the two pan Arctic domains. The variation is smaller in the global domain as Antarctica has the maximum number of active points when Arctic is at its minimum and vice versa. These variations can impact the strategy for allocation of memory as one goal is to reduce the memory usage.





| Domain | total | water | ice winter | | ice summer | |
|---|---|---|---|---|---|---|
| | | | $T \cap U$ | $T \cup U$ | $T \cap U$ | $T \cup U$ |
| DMI-NAAg | 1,662,465 | 1,000,954 | 606,797 | 624,830 | 290,171 | 299,148 |
| RASM | 14,745,600 | 9,314,922 | 2,762,746 | 2,822,197 | 1,510,341 | 1,549,195 |
| gx1 | 122,880 | 86,354 | 10,182 | 11,479 | 10,782 | 11,515 |

**Table 1.** Number of points for the four test domains: the total number of grid points excluding the boundary condition, the number of water points, the number of active $T$ and $U$ points and the number of active $T$ or $U$ in winter and summer.

## 2.4 Test setup

The results in this study are primarily based on a standalone test using inputs that have been taken from realistic CICE runs before and after the subcycling. The refactorizations have been tested in three stages (V0, V1 and V2, as described below).

**V0** Standard EVP solver.

**V1** Single core refactoring of the memory access patterns used in the EVP solver

**V2** Single node refactoring illustrating four different OpenMP approaches and one pure Fortran 2018 `do-concurrent` approach, including conversion from pre-computed grid arrays to scalars recomputed at every iteration.

Unit tests have been conducted with different sets of compiler flag optimizations from very conservative to very aggressive. A weak-scaling feature has also been added to the standalone test in order to measure the performance at different resolutions/number of grid points and to allow for full node performance tests.

The refactorization and its impact on performance has been tested on four types of architectures (table 2). The goal is to demonstrate the effect of the bandwidth limitation and the performance enhancement on CPUs and GPUs. All CPU executables were built using the Intel® Classic Fortran compiler and all the GPU executables were built with the Intel® Fortran compiler from the oneAPI HPC toolkit 2023.0.

| Name | Type |
|---|---|
| 3rd Gen Intel® Xeon® Scalable Processor | 72core-CPU+DDR4 memory |
| 4th Gen Intel® Xeon® Scalable Processor | 112core-CPU+DDR5 memory |
| Intel® Xeon® CPU Max Series | 112core-CPU+HBM memory |
| Intel® Data Center GPU Max Series | GPU+HBM memory |

**Table 2.** Description of the CPUs and the GPU used in this study. Hardware listed with HBM includes high bandwidth memory. More information can be found on https://www.intel.com/content/www/us/en/products/overview.html.





All performance numbers reported are the average time obtained when repeating the test ten times. All timings are obtained using `omp_get_wtime()`. For the capacity measurements, 8 ensemble members run the same workload simultaneously evenly split across the full node/device/set-of-devices. The timing of an ensemble run is the time of the slowest ensemble
member, and we repeat the ensemble runs ten times and report the average. All performance experiments on 4th Gen Intel® Xeon® Scalable Processor and Intel® Xeon® CPU Max Series are done in SNC4 mode and with HBM-only on Intel® Xeon® CPU Max Series.

The GPU results indicate only the compute part and not the usually time-consuming data-traffic between the CPU and the GPU. Because the kernel constitutes a single model time-step, most data traffic in this kernel is one-time initialization and
hence would not contribute to the compute time in the $N - 1$ remaining time-steps of a full simulation.

Finally, the refactorizations are tested in CICE (Hunke et al., 2023b) with different compiler optimization flags. This initial integration is focused solely on correctness; section 4 presents our proposal for a performance focused integration. The performance focused integration can be considered a refactoring at the *cluster-level*, on top of the refactoring at the *core-* and *node-level* reported here.

The new method does not include any new physics. Therefore, it is important that the results remain the same. This is verified by checking that restart output files contain "bit-for-bit" identical results at the end of two parallel simulations. This requires identical *md5sum*s on non-optimized code. All tests verify this (not shown).

When the build uses more aggressive compiler optimization flags, the binary executable may use operations that produce a different final round-off error, or that do the exact same calculations in another order resulting in bit-for-bit differences due
to the discrete representation of real numbers. For instance, a fused multiply-add operation has 1 rounding instead of one multiplication followed by an addition requiring 2 rounding operations. Such a deviation does not originate from differences in the semantics dictated by the source code itself, which expresses exactly the same set of computations. The difference originates from the ability of the compiler to choose from a larger set of instructions. It is important that the numeric should be stable in that respect. For the optimized, non-bit-for-bit runs, we verified that the numerics are stable to this kind of unavoidable
numerical noise using a Quality Control (QC) module (Roberts et al., 2018) provided with the CICE software.The CICE QC test checks that two non-identical results are statistically the same with respect to climate based on a five year simulation with the domain gx1. The QC test compare the sea ice cover and the sea ice thickness. The result is shown in figure 2.

The implementation passes the QC test for integration into CICE as the noise level is lower than the criterion.

## 3   Performance results

There are several ways to measure and evaluate compute performance. This section focuses on evaluating the EVP standalone kernel performance as measured by *time-to-solution* on different compute nodes. The evaluation is split into two steps, single-core performance and single-node performance.

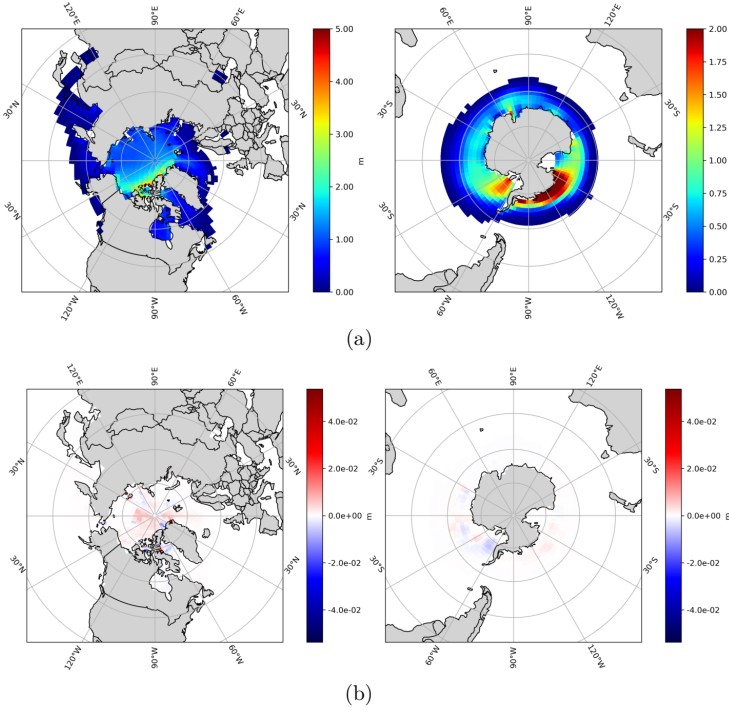

**Figure 2.** a) Sea ice thickness in the northern and the southern hemispheres on 31 January 2009, after 5 years of simulation starting on 1 January 2005 and using the gx1 grid provided by the CICE Consortium (b) Difference between the standard EVP solver and the EVP solver described in this study on 31 January 2009.

## 3.1 Single core performance

The motivation for the single core refactorization described in section 2.2.1 was to allow the compiler to utilize vector (SIMD)
instructions instead of confining itself to x86 scalar instructions for both memory accesses and math operations. The instruction sets formally known as AVX-2 (256 bit) and AVX-512 (512 bit) constitutes two newer generations of such vector (SIMD) instruction extensions to the x86 instruction set architecture for microprocessors from Intel and AMD. Compiler options can be used to specify specific versions of SIMD instructions but the compiler can only honor this request if the code itself is SIMD vectorizable.

Figure 3 shows the single core performance of the different EVP implementations described in listings 2-3 and listing 5-9 for domain RASM described in table 1.

Figure 3 shows limited improvement within the upstream implementation (`v0-base`) when applying either AVX-2 or AVX-512 as described in section 2. The improvement factor of the single core refactorization is approximately 1.6 when SIMD instructions are not used in the compiler instructions, 3.8 when AVX2 is used and 5.1 when aiming at AVX512 code
generation. Moreover, the SIMD improvements are achieved across the different OpenMP versions. Although all the refactored



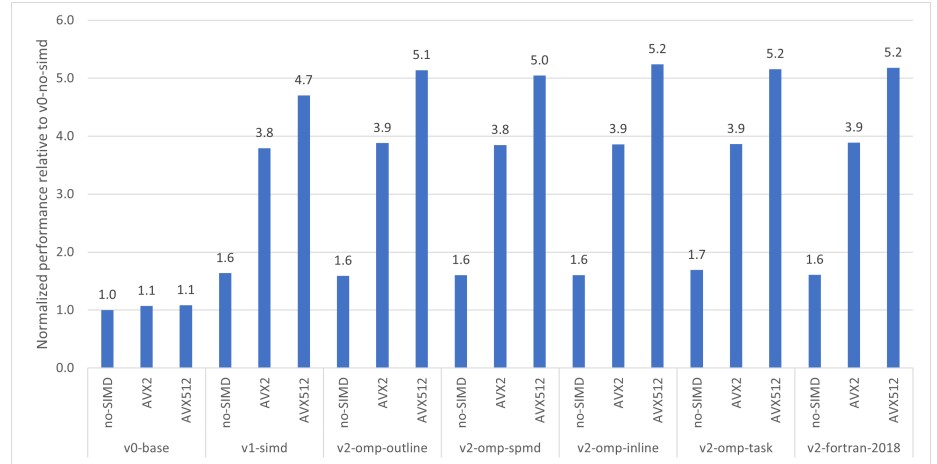

**Figure 3.** Single core 3rd Gen Intel® Xeon® Scalable Processor (8360y) performance for the same algorithm (EVP) implemented via different approaches and with the build process requesting no-SIMD, AVX2 and AVX512 code generated, respectively. The prefix versions v0, v1 and v2 are defined in section 2.4 and the baseline is the original implementation (v0-base) without SIMD generation. Each bar show the improvement factor compared to the baseline.

versions show the same performance, this is not given a priori, since the intermediate code representation given to the compiler back-end is expected to be different for each of these representations.

The single core refactorization improves the code generation and the associated performance. In addition, the one-dimensional compressed memory footprint is much more efficient that the standard two-dimensional block structure as it reduces the memory footprint by the ratio of ice points versus grid points. For the RASM case, it amounts to a reduction factor of 5 in winter and 10 in the summer (see table 1). Importantly, all points in the 2D arrays used in the standard implementation have to be allocated, whereas the refactored data structure only needs the active points allocated.

### 3.2  Single node performance

Efficient node performance requires that an implementation have both good single-core performance and good scaling properties. The main target of this section is to describe the performance results as measured by the *time-to-solution* on a given node architecture. The performance diagrams found in this section show the performance outcome when all the cores available on the node/device are used. The refactored code is ported to GPUs using OpenMP target offloading. For the capacity scaling study this allow us to run on hosts that support multiple GPU devices but for the strong scaling study we have confined ourselves to the use of classical OpenMP offloading, which currently only supports single device offloading, cf. (Raul Torres and Teruel, 2022).

In addition, single node performance is measured according to the relevant hardware metrics. Because the EVP implementation is memory-bandwidth bound, it is relevant to compare the sustained bandwidth performance of EVP with the well-established bandwidth benchmark STREAM triad, cf. McCalpin (1995) and figure 4. The STREAM triad benchmark delivers

a main memory bandwidth number measured in Gb/s and is considered to be the practical limit sustainable on the system being

measured.

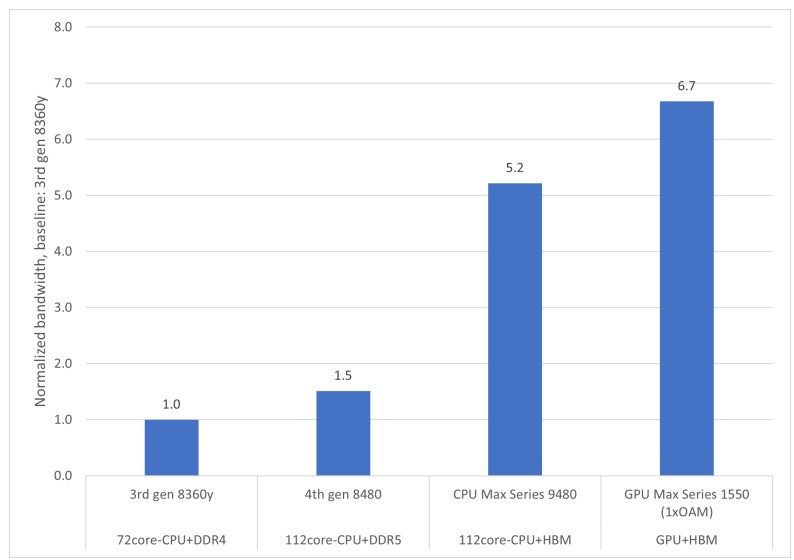

**Figure 4.** This figure shows the improvement factor for STREAM triad memory bandwidth benchmark and show us what is achievable at best for solely bandwidth bound code. First bar is the baseline based on  3rd Gen Intel® Xeon® Scalable Processor, second bar is  4th Gen Intel® Xeon® Scalable Processor, third bar is  Intel® Xeon® CPU Max Series  and the fourth is  Intel® Data Center GPU Max Series, cf. table 2.

Section 3.2.1 focuses on performance results for a strong scaling study whereas section 3.2.2 focuses on capacity scaling. Single-node performance is evaluated on the architectures described in table 2 and baseline performance is always such that the measured bandwidth of EVP coincides with that of STREAM triad on the node.

### 3.2.1   Strong scaling - OpenMP and OpenMP target

*Strong scaling* is defined as the ability to run the same workload faster by using more resources: the ability to strong scale any workload is governed by Amdahls law (Amdahl, 1967).

Figure 5(a) show the impact of the choice of architecture on the single-node performance results on the four architectures described in table 2 for the refactored EVP. Note that  3rd Gen Intel® Xeon® Scalable Processor  only has 72 cores.

The improvement factor between the CPUs with DDR-based memory coincides with the improvement factor obtained by

STREAM triad (figure 4), which is considered the practical achievable limit of the hardware. The improvement factor for the two bandwidth optimized architectures ( Intel® Xeon® CPU Max Series  and Intel® Data Center GPU Max Series) is less than the corresponding improvement factor obtained by STREAM triad. This shows that bandwidth to memory is no longer the limiting performance factor. This finding will be discussed further in section 4.



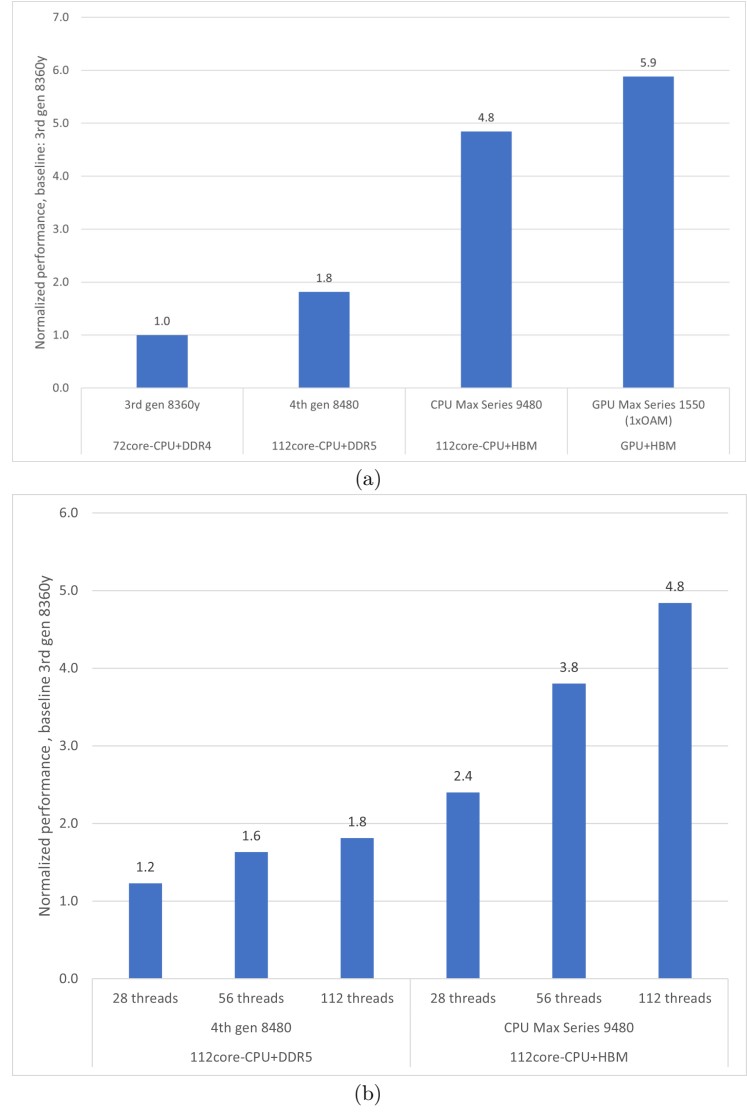

(a)

(b)

**Figure 5.** The improvement factor for the refactored EVP. The baseline for a) and b) is the performance of 3rd Gen Intel® Xeon® Scalable Processorwhen using all cores. a) Strong scaling performance when using all available cores on one node. First bar is 3rd Gen Intel® Xeon® Scalable Processor, second bar is 4th Gen Intel® Xeon® Scalable Processor, third bar is Intel® Xeon® CPU Max Series and the fourth is Intel® Data Center GPU Max Series, cf. table 2. on different architectures, cf. table 2. b) Strong scaling performance at three different core counts for 4th Gen Intel® Xeon® Scalable Processorand Intel® Xeon® CPU Max Series .

Figure 5(b) show the performance at different core counts for the two hardware types ( 4th Gen Intel® Xeon® Scalable Processor and Intel® Xeon® CPU Max Series ) that are similar except for their bandwidth. The first observation is that the performance of the HBM based CPU is better. The second observation is that the DDR based hardware stops improving





the performance at approximately half the number of cores available on the node, preventing further scaling on that memory system. With the HBM hardware the code scales out to all the cores on the node. The improvement factor differs because the sustained bandwidth becomes saturated on the 4th Gen Intel® Xeon® Scalable Processor memory. This underlines the

importance of refactoring of code in order to reduce the pressure on the bottleneck, which in this case is the bandwidth. It also demonstrates that the hardware sets the limits for the potential optimization.

If both the standard and the new implementations of the EVP solver strongly scales equally well, then the node improvement factor should be the same as the single core improvement factor found in section 3.1. The observed improvement factors of refactored versus standard EVP on 4th Gen Intel® Xeon® Scalable Processor and Intel® Xeon® CPU Max Series are 13

and 35 (not shown), respectively, i.e. the new EVP solver also scales better than the original EVP implementation on both systems. The standard EVP allow for multiple decomposition's, thus other decomposition's may affect the result, however the conclusion remain the same.

### 3.2.2 Capacity scaling - OpenMP and OpenMP target

*Capacity scaling* is defined as the ability to run the same workload in multiple incarnations (called ensemble members) si-

multaneously on multiple compute resources. *Perfect capacity scaling* is achieved when we can run $N$ ensemble members on $N$ compute resources, with a performance degradation bounded by the *run-to-run-variance* measured when running one ensemble member on 1 compute resource and leaving the rest of the compute resources idle. This performance metric indicates how sensitive the performance is to what is being executed on the neighboring compute resources. We find perfect scaling on the DDR-based systems, i.e. the variance of the timings between individual ensemble members are similar to the variation in

timings of repeated single member runs.

Figure 6 summarizes the capacity scaling results.

The improvement factor between the two DDR-based systems coincides again with the improvement factor obtained by STREAM triad, which means that the performance is bandwidth bound. The improvement factor for the bandwidth optimized architectures is somewhat less than the corresponding improvement factor obtained by STREAM triad. This is discussed further

in section 4.

HPC systems with GPUs typically host multiple devices per node and this is the reason that we have conducted a multi-device experiment. The experiment shows that the performance continue to increase, however the energy required to drive a node with 4 GPU devices is significantly higher than the energy required to drive the dual-sockets CPU that we cross-compare with. The energy budget has not been highlighted in this study.

### 435 4 Discussion

These results demonstrates a refactorization of the EVP solver within CICE that takes full advantage of modern CPUs and GPUs. Moreover, the new implementation is easy to adapt to an unstructured grid, although it is implemented here on a




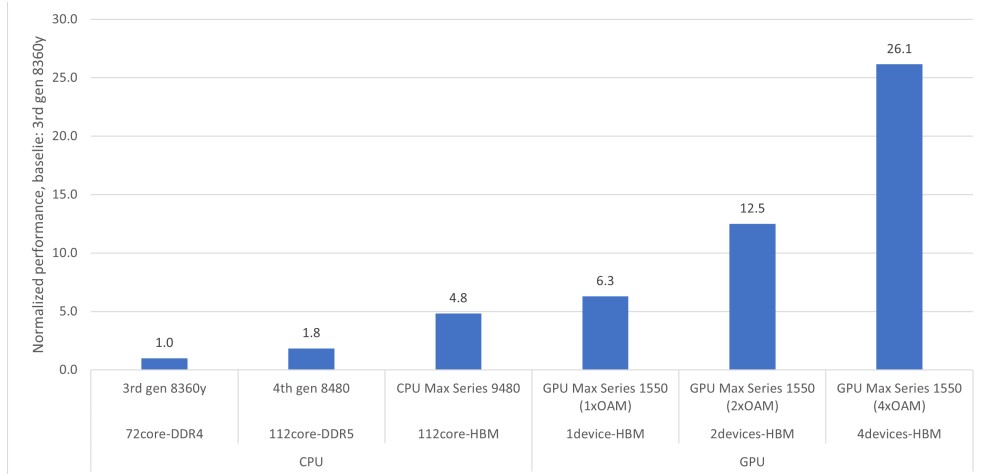

**Figure 6.** Normalized performance for capacity scaling of the refactored EVP using the 3rd Gen Intel® Xeon® Scalable Processor (3rd gen 8360y) node as the baseline, and all other results normalized relative to this performance. The baseline sustains the same bandwidth as STREAM triad. The figure also illustrates how much 1, 2 and 4 GPU devices per node matter to the collected node throughput.

structured grid. That said, there were choices both in the refactorization of the EVP solver and within the integration into CICE, which are discussed in this section.

Both the new single-core SIMD parallel performance and the new single-node OpenMP parallel performance are evaluated in the previous sections. The refactored code reaches the peak bandwidth performance of the two DDR-based memory systems, but the peak bandwidth of the HBM based system is not reached. On the Intel® Xeon® CPU Max Series with HBM and running in HBM-only mode, we reached $\approx 80\%$ of the practical peak bandwidth. This result is consistent for both capacity scaling and strong scaling. Although the improvement factor is the same for both types of scaling, the reasons for the performance gap, as

compared with the theoretical limit defined by the STREAM triad, are quite different.

The issue for the capacity scaling is that it enforces lower frequency (both core and uncore) when all NUMA domains operate simultaneously, compared with operating a single NUMA domain. Computations are slower (drop in core frequency) when all NUMA domains are in use simultaneously and we have higher memory latency (drop in uncore frequency). Combined, they lead to slower overall execution. This also shows that using STREAM triad as our performance proxy is too simplistic on the

bandwidth optimized CPU. There is a limit to the number of floating point operations that one can put into the STREAM triad and retain the base frequencies of the core and uncore. On the other hand, the strong scaling gap for the Intel® Xeon® CPU Max Series CPU is solely due to limitations in the algorithm and data set, where there is an inherent imbalance.

The first integration step described in this study used the current infrastructure within CICE and focuses solely on correctness, not on the performance, in order to establish a solid foundation for future work. For instance, the implementation utilizes

existing gather/scatter methods to convert some of the arrays from 1D to 2D global to 3D block-structure, which is used for parallelization in the rest of the CICE code, and vice versa. It would be better to convert the 1D arrays directly to the block-





structure (1D to 3D). The number of calls to gather/scatter methods could also be reduced. The ideal solution would be for all spatial quantities to only exist as 1D vectors; most EVP loops are already geared towards this as they loop in 1D space with pointers to the 2 indices used in the array allocation. The halo-swaps can be re-introduced directly on the new 1D data
structures using `MPI-3 neighborhood collectives`.

A major performance challenge within the standard EVP solver is the halo updates required at every EVP subcycle as each halo update introduces a MPI synchronization. Better convergence is achieved when the number of subcycles increases, but this also increases the number of MPI synchronizations linearly. As a consequence of this the inherent imbalance challenges on irregular data increases. The goal is improve performance of the full model, therefore the number of synchronizations must
be reduced.

The initial integration only includes MPI synchronizations at the time step level but does not allow CICE to be split into sub components as it is suggested below. Therefore it confines itself to use the same number of threads for EVP as for the rest of the CICE components and EVP will consequently only be able to utilize a single MPI task and leave the remaining MPI tasks idle. This is obviously a very inefficient integration as it retain the observed scaling challenge at the *cluster*-level.

To improve the initial integration, performance and to cope with the underlying challenges, an alternative approach is suggested leveraging MPMD parallelization (Mattson et al., 2005). This allows heterogeneous configurations, where the EVP solver could run on separate hardware resources and/or utilize different parallelization strategies (e.g. pure MPI, hybrid OpenMP-MPI, hybrid OpenMP-MPI running OpenMP offloading) relative to the rest of the model. If a time lag is implemented between the two components, they could run concurrently. This approach is also beneficial for the performance of the
rest of the model, as it relieves the model from carrying EVP state variables and prevents flushing the caches with EVP state at every time step. It will also allow runs of several EVP ensemble members on a single node serving a set of model ensemble members each running on their own set of nodes. At last, it is easier to integrate the new EVP component into other modeling systems, because it will have a pure MPI interface.

The MPMD pattern is generally used by ESM communities to handle coupled model systems, see e.g. where the ocean and
the sea ice model run on different groups of the cores/nodes see e.g. (Ponsoni et al. (2023); Craig et al. (2012)). Sometimes it is also used internally within systems for I/O, see e.g. the ocean model NEMO (Madec et al., 2023). To the best of our knowledge it is *not* common to see the MPMD pattern for model sub-components beyond IO handling nor is it common that it supports heterogeneous systems.

This new implementation of the EVP solver includes a strategy for how to allocate data. The strategy selected for this
integration is to allocate all ocean points and then check whether or not they are active within the calculations. For the domains in this study this produces a large overhead, since there are many ocean points that are never active (see table 1). However this behavior is domain specific and very different for different setups. A second strategy could be to reallocate all 1D vectors at every timestep and then only allocate the active point. This includes an overhead for reallocating at every time step, but it reduces the memory usage. An alternative, in-between method would be to only reallocate when the number of active points
increases above what has been allocated. This last strategy is the one we propose for the final MPMD based MPI refactoring in CICE.





## 5    Conclusions

This study analyzed the performance of the EVP solver in the sea ice model (CICE) and found performance challenges with the standard parallelization at the *core-*, *node-level* and *cluster-level*. As a consequence of this a refactorization of the solver is
developed. The evaluation shows that it is possible to obtain significant performance and memory footprint improvements.

The refactored EVP improved the performance with a factor of 5 when compared to the original version when 1 core is used on 3rd Gen Intel® Xeon® Scalable Processor. This improvement is primarily caused by a change in the memory access patterns from random to direct. When using 112 cores (full node) the improvement factor on the 4th Gen Intel® Xeon® Scalable Processor  is 13 and on the Intel® Xeon® CPU Max Series  is 35. The study showed that the limiting performance factor for
EVP on traditional CPU's is the memory bandwidth. This is the main difference between the two types of hardware and the main reason for the difference in performance on a full node.

The refactored version was capable of sustaining STREAM triad bandwidth (practical peak performance) on the CPUs within this study that are based on DDR-based memory. For strong scaling on the Intel® Xeon® CPU Max Series  only 80% of the bandwidth was used due to imbalances in the algorithm and the datasets. At last GPUs deliver higher memory bandwidth
than CPUs so we also ported the new implementation to run on nodes with GPU devices.

The single-node improvements are merged back into the CICE model with a focus on the correctness. Next step will be to improve the integration with a focus on full model performance on both CPU's and GPU's.

All CPU and GPU performance was achieved solely by using open standards, OpenMP and oneAPI in particular.

*Code and data availability.*  The code for CICE v6.5 for the QC test can be found https://zenodo.org/records/10056499. This version do not
include the integration. This is available in the most recent release. Data sets for this can be found https://github.com/CICE-Consortium/CICE/wiki/CICE-Input-Data with DOI numbers: 10.5281/zenodo.5208241, 10.5281/zenodo.8118062, 10.5281/zenodo.3728599.

Unit test for the EVP solvers can be found Rasmussen et al. (2024). Data sets are available upon request.

## Appendix A:  Abbreviations

*Author contributions.*  JP contributed with the main idea of the re-factorization of the EVP with input from TR, MR and RS. TR integrated
the refactored EVP with support from JP, MR AC, SR and EH. TR and JP wrote the manuscript with input from all others.

*Competing interests.*  To the authors knowledge there are no competing interest.





**Table A1.** Acronyms

| Abbreviation | Full name |
| --- | --- |
| CICE | The Los Alamos sea ice model |
| DMI | Danish Meteorological Institute |
| DDR | Double Data Rate |
| ESM | Earth System Model |
| EVP | Elastic-Viscous-Plastic |
| HBM | High Bandwith Memory |
| MPMD | Multiple Program Multiple Data |
| NUMA | None-Uniform memory access |
| QC | Quality Control |
| RASM | Regional Arctic System Model |
| SIMD | Single Instruction Multiple Data |
| SPMD | Single Program Multiple Data |
| VP | Viscous-Plastic |

*Acknowledgements.* The study is funded by the Danish State through the National Centre for Climate Research (NCKF) and the Nordic council of Ministers through the NOCOS DT project. Elizabeth C. Hunke was supported by the U.S. Department of Energy Office of Biological and Environmental Research, Earth System Model Development program. Anthony P. Craig was funded through a National
Oceanic and Atmospheric Administration contract in support of the CICE Consortium.





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
