# Peer review of "Refactoring the EVP solver from the sea ice model CICE v6.5.1 for improved performance"

_Geoscientific Model Development, 2024_

## Author Response (AR1)

**We would like to thank the reviewers for their comments. Responses to the comments are marked in bold below. We have added relevant information to the manuscript and clarified where needed based on the reviewers' comments and questions.**

*Review 1*

All the performance results are normalized to the reference implementation. Although the comparison to the stream benchmark makes the argument, convincing about the memory bandwidth optimization a few absolute numbers in terms of achieved memory bandwidth vs theoretical could make the paper of broader interest. If some performance metric numbers were available, it would be good to provide some, or show some of the optimizations on roofline model plot.

 - The paper is only considering Intel Hardware. This is ok, also considering that one of the Author is from Intel, nonetheless further comments should be made on applicability to other hardware, in particular Nvidia or AMD GPUs since this may be more broadly available in the ESM HPC community. There is reference to one-API it is not clear from the paper if the openmp target is using some proprietary extension or if this should work out of the box on other hardware.

**All implementations only use open standards. There are no proprietary extensions used. Therefore, it is expected that similar results can be achieved on NVIDIA or AMD hardware, for example.**

**Comment is added on line 330**

- l.369 : "3.8 when AVX2 is used and 5.1 when aiming at AVX512" most of the argumentation is that performance increased is gained because of better memory usage. It is therefore not clear why there is such a gain from vectorization, this would need a comment

**SIMD is a matter of both COMPUTE instructions and MEMORY instructions, and it requires AVX2 or AVX512 instructions to load/store instructions to utilize the High Bandwidth Memory. The 2D arrays have random memory access, whereas the 1D arrays have direct memory access.**

**Section 2.2.1 (lines 115 – 122) addresses this. Additional comment is added to lines 370 and 380 - 385.**

Technical corrections

- p1. one could say in the abstract and maybe in the title that CICE is a sea ice model

**Sea ice model is mentioned in the abstract and the title**

 - l.446 : "it enforces lower frequency" : maybe comment if this is a strict hardware requirement, or could this be deactivated ?

**This is a general hardware feature that ensures the hardware does not overheat. It cannot be deactivated.**

**Comment added to lines 457 - 462.**

**For Intel this is called RAPL. See e.g. https://www.intel.com/content/www/us/en/developer/articles/technical/software-security-guidance/advisory-guidance/running-average-power-limit-energy-reporting.html**

- l.464: "The goal is improve performance of the full model " -> "The goal is to improve performance of the full model "

**Corrected**

*Review 2*

**A general comment: the prime objective of this manuscript is to describe the performance of an EVP unit test, not the full CICE model. The aim of the implementation within CICE is to demonstrate that it is functional, not to show "perfect" performance. Improved performance in the full system requires handling of MPI, for example, as is noted by the reviewer. Performance of the full CICE model is beyond the scope of this manuscript.**

**Comment is added to the abstract and lines 314, 344 and 462.**

This paper analyze the performance of Elastic-Viscous-Plastic (EVP) solver from the CICE model. The EVP solver was extracted and refactored in order to improve the performance through the change of memory patterns from random to direct. The speed up is 5 times on 3rd Gen Intel Xeon Scalable processor, 13 times on the 4th Gen Intel Xeon processor and 35 times on the Intel Xeon CPU Max Series due to the memory bandwidth differences. Improved performance can also be seen on GPU processors. This is a good advance for the EVP solver within CICE model development. However, there are some major issues to be addressed.

This paper tests the CPU and GPU together (Fig. 6). Numerical accuracy is very important for the EVP solver. Is this a fair comparison?

**It is true that performance results are shown for both CPUs and GPUs. The aim is to show that the scalability/performance can be achieved on both types of processors, especially with high bandwidth CPUs, which are somewhat similar to GPUs.**

**Comparing different hardware types to each other is not entirely fair as there are other issues that affect the choice of hardware. This could be price or energy consumption.**

**A comment has been added to line 451.**

Particularly, ESM is very sensitive to the CICE. Line 340-342 mentioned that "bit-for-bit" identical results are achieved in the CICE (or just the algorithm itself). Can the authors confirm that identical results are achieved when they compare the performance, even for the GPU computation?

**Bit-for-bit reproducibility is discussed for the EVP kernel as a unit test for all 7 different implementations on the CPU. Small changes to the results are expected when hardware is changed (e.g. CPU to GPU). It is not possible to run the baseline test (v0) on GPUs, thus it is not possible to make a bit for bit comparison for GPUs. Comment added to line 350.**

**The aim of the bit-for-bit test of the EVP unit test is to verify the correctness of the EVP refactorization. First, it ensures that the changes to the Fortran code and the data structures used do not change the results. Second, it ensures that the threading introduced in the code does not change the results. This paper focuses on the EVP algorithm, but the considerations will be the same when bit-for-bit is discussed for the full CICE model.**

**The algorithm is bit-for-bit with conservative flags and no optimization. When optimization flags are used, bit-for-bit results can not be guaranteed, but the solution should not change significantly. The purpose of the five-year run with CICE shown in figure 2 is to demonstrate that the solution does not change significantly. Although the implementation into CICE is not the focus, it was necessary in order to show that the full model was still functioning correctly with the new algorithm. Optimization of the CICE implementation is beyond the scope of this manuscript. Efficiency of the integration will be our next step, including simulations on multiple nodes.**

That means you also use double-precision? Did I miss something? If not, can the authors provide the accuracy level for double precision in CICE6? Is the performance affected by the accuracy used?

**This study is confined to double precision (as in the rest of CICE: selected_real_kind(13)). Performance is affected by the choice of accuracy (single precision vs double precision); lower accuracy would improve the performance as it puts less pressure on the bandwidth, but it will be important to check that no significant changes are seen in the results. For this reason, some variables may have to be kept as double precision at all times. No additional comments has been added.**

Also, this study only consider the Intel processes. What about the other processes? If the memory bandwidth is the only concern here, the brand-dependence should be small. The authors should address and discuss these issues to ensure the robustness of the proposed change.

**It is correct that the brand or choice of the CPU influences the performance effect. Performance measurements are confined to the hardware and software that is tested (see figure 4). For instance, this study compares two types of hardware where bandwidth is the only difference. This changes performance significantly. None of the changes of the source code is specific to the hardware, and the relative performance of stream Triad on a given CPU will be a good proxy to estimate the performance effect of the refactoring on this CPU.**

**Comment line 330 about the brand of the software.**

This paper extracts the EVP solver out from CICE6. The major improvement is done only within the single processor. Is this practical for the real application?

**It is assumed that the reviewer means a single node with multiple processors. We consider this approach to be common (e.g. the ESCAPE projects at ECMWF). It is also efficient to focus on one part of the code and its refactorization. Having said that, it is indeed a valid point that once the refactoring is completed there will be another phase that focuses on improving the integration into CICE. As mentioned, this is beyond the scope of this manuscript, but the discussion includes some thoughts on this topic.**

**See also the comment at the beginning of this review section.**

Particularly, CICE6 is currently using a structured grid. The standard procedure uses indxti(ij) in order to avoid the land points or unused points. Is the refactored algorithm in Listing 3 and 4 effective?

For example, if your ee(iw), iw and iw+1 are not pointing to the neighboring points in the array. Also, the size of ee is change at every time step when considering this. Are you sure your algorithm will be faster? These real applications should be discussed.

**Yes, the refactored algorithms are indeed effective and significantly faster, as we illustrate with the performance measurements. The stencil operations will point to non-contiguous memory cells, but this is handled with gather operations, and reading the data from neighboring cells is not a dominant factor. We quantify the gather reads relatively in the paper to address this concern. Also, note that there are no write operations into neighboring cells.**

**Line 112 – 120 in the revised manuscript (no changes) describes this problem in version 0 (2d), which results in random memory access that prevents vectorization. The 1D version is mostly direct memory access, which allows for vectorization. This is the reason why the refactored version is faster for both structured and unstructured grids.**

This paper only shows the single node performance. However, CICE multi-domain MPI. In practice, CICE uses hundreds of processes. Some discussion should be included on the performance if multi-nodes are used.

**Multiple-node performance is out of scope and will be addressed during the upcoming integration phase. The manuscript includes discussion of how we intend to integrate the refactored code properly performance-wise, and only after that point will it make sense to do a multi-node performance study. A brief overview of the next step towards this is described from line 490 to 498.**

What about the land or non-active points at different time?

These points are changed with time, right? Do you consider the additional time tracking these changes? What about the load balance among different nodes/cores?

**The conversion back and forth from 2D is not included in the timings. In general the 2D grid is an abstraction for the computer. Ideally the full code would use 1D arrays. Land points are always eliminated in the 1D solver and are not tracked. Different approaches can be made to distinguish between active (ice covered cells) and non-active, ice-free cells. One can allocate all water points, or only active points. If the number of active points increases above a maximum limit, one can reallocate. The most efficient method depends on the domain. Load balancing is tricky for sea ice models in both 1D and in 2D. The 1D vector is ideal for OpenMP as there are fewer inactive points. Load balancing for MPI is our next step, which we discuss in the manuscript.**

**Added comment on lines 310**

Can the author address the overall performance improvements of CICE based on this new refactorization? Or is the performance test based on the 5 year simulation in CICE gx1 domain? There is no discussion how the performance results are obtained. What's the model set up?

Is this using the standard CICE example? The performance improvement in the global CICE example should be demonstrated.I suggest to give the timing how much this can save the standard CICE gx1 overall simulation time (e.g., 5 year simulation in Fig. 2).

**For this paper, performance testing is based on a unit test of the EVP solver. The gx1 model is relatively small and therefore not ideal for performance testing in realistic cases. The RASM and the DMI model setups are used, see figure 1. These are both regional domains but for this purpose, to put pressure on the system, there are more active points.**

**The purpose of the 5-year CICE simulation is only to show that integration of the optimized algorithm into CICE fulfills the requirement that it not cause large changes in the model simulation. See line 294**

Finally, there are some grammatical errors within the text. The text and discussion also require some reorganization and connection for a better presentation. Some paragraphs have only 1 sentence, e.g., line 365, 395, 426, 508. Is this on purpose?

**We have reorganized the manuscript where needed in order to avoid this.**

Further improvement in the English and careful proofread by a native speaker are required. This paper is appropriate to be published in GMD after considering the above major issues and the following comments.

1.Line 50-54 discuss the number of active sea ice points varies in time. The algorithm doesn't consider this overhead in CICE. Can the author discuss how that can affect the generated 1D array of ee in real time?

**None of our performance tests include the variation in active sea ice points within CICE. This is beyond the scope of the paper, since a proper MPI strategy must be implemented first. The generated 1D vectors (including ee) will all have the same length. This is discussed in lines 494 and onwards.**

2.Line 115, missing word after "Reinders and". Please check the wording throughout the manuscript.

**Corrected in bibtex file**

3.Section 3.2.2, the title of this section is "Capacity scaling-OpenMP and OpenMP target", however, I didn't see any OpenMP is discussed here.

**The heading has been changed to match the content in both sections, strong scaling (3.2.1) and capacity scaling (3.2.2).**

4.Line 437, I do agree that this approach is ideal for unstructured grid. However, CICE is still based on structured grid. Some comments should be included.

**The computer only sees data as 1D arrays in memory regardless of the data structure. The important part here is that memory access is mostly changed from random to direct. See for instance line 384.**

5.Line 446-450, these paragraphs are unclear. What do you mean by "Combined, they lead to slower overall execution".

**Updated lines 459 to 464**

6.Line 450, floating point operations are discussed here. Do you use the same floating point operations in CPU and GPU configuration?

**Yes, the Fortran code within the OpenMP scope is the same for both CPU and GPU. The only difference is the OpenMP directives.**

Line 457-460 discusses a major issue about the cross-node communication. In the structured grid, we can easily define the halo-region. However, the current change seems to be more ideal for the unstructured grid arrangement and parallelization. The improvement on a real CICE application can be better demonstrated.

**It is beyond the scope of the manuscript to optimize MPI in CICE. This is for the next phase. That being said, it is not optimal to have hundreds of MPI communications per time step and limited calculations within the EVP solver. This is the reason why OpenMP is by far the preferred communication method on one node. The current (2D) OpenMP threads are based on the MPI blocks and still need to update halo's. The 1d version do not need to communicate as the threading share memory when using OpenMP. Obviously it is not ideal to limit the solver to one compute node. This part is discussed in the discussion and future work will focus on this.**

7.Line 479-483, this section discusses the MPMD parallelization. However, it is not clear to me how this can be implemented within ESM and CICE. How this discussion connects to this study?

**The aim of MPMD is to run with local MPI configurations within the different parts of CICE or the ESM. This will allow each part of the code to be executed on a limited number of nodes. This simplifies the communication and makes it cheaper. We have attempted to clarify this point in the text.**

8.Line 484-491, this discussion is very important. The overhead in the selected strategy needs to be addressed. Several potential strategies are mentioned but not fully addressed. We can see the refactorization can enhance the single node performance. However, is this approach ideal for a real CICE application? The improvement in the global CICE model should be demonstrated.

**It is beyond the scope to demonstrate the full performance of CICE. We have clarified this point in the text. Therefore the discussion does not draw conclusions regarding the integration approach. See also the first comment to this review.**